



# Interpretation of zircon corona textures from metapelitic granulites of Ivrea-Verbano Zone, Northern Italy: Two-stage decomposition of Fe-Ti oxides

Elizaveta Kovaleva[1,2], Håkon Austrheim[3], Urs Klötzli[2]

[1]Department of Geology, University of the Free State, Bloemfontein, 9300, 205 Nelson Mandela Drive, Free State, South Africa

[2]Department of Lithospheric Research, Faculty of Earth Sciences, Geography and Astronomy, University of Vienna, Vienna, A-1090, Althanstrasse 14, Austria

[3]Section of Physics of Geological processes, Department of Geoscience, University of Oslo, Oslo, 0316, Norway

*Correspondence to*: Elizaveta Kovaleva (kovalevae@ufs.ac.za)

**Abstract.** In this study we report the occurrence of zircon corona textures in metapelitic granulites of the Ivrea-Verbano Zone. Unusual zircon textures are spatially associated with Fe-Ti oxides and occur as (1) vermicular-shaped aggregates 50-200 μm long and 5-20 μm thick, and as (2) zircon coronas and fine-grain chains, hundreds of μm long and ≤1 μm thick,

spatially associated with (1). Formation of such textures is a result of mineral-fluid reactions, which occurred in two stages and involved: (1) decomposition of ilmenite to Zr-rich rutile and vermicular-shaped zircon during peak metamorphism and initial cooling stage, and (2) recrystallization of Zr-rich rutile to Zr-depleted rutile and submicron-thick zircon coronas during further exhumation and cooling. We also observed hat-shaped grains that are composed of preexisting zircon overgrown by zircon coronas during stage (2). Hat-shaped grains have a flat surface towards the oxide phase, which indicates partial

dissolution of preexisting zircon grain. Formation of vermicular zircon (1) preceded ductile and brittle deformation of the host rock, as vermicular zircon is found both plastically- and cataclastically-deformed. Formation of thin zircon coronas (2) was coeval with or soon after the brittle deformation, as coronas occasionally fill fractures in the host rock.

Occurrences of zircon coronas has important implications in fundamental studies regarding metamorphism, metasomatism and element transport in the Earth's crust. We demonstrate that metamorphic zircon can nucleate and grow as a result of

hydration reactions at the cooling stage after granulite-facies metamorphism, and reflects stages of rock evolution. Zircon corona textures are the tool for indicating metamorphic and metasomatic reactions in the host rock, and establish the directions of the reaction front.

## 1 Introduction

### 1.1 Occurrences of fine-grain zircon and zircon corona textures in metamorphic rocks

Zircon occurs as an accessory mineral in many rock types and usually forms euhedral elongated single crystals that are shaped by a combination of prismatic and pyramidal faces (Corfu et al., 2003). Rarely zircon can appear in other shapes, such as micro-granular, needle-shaped or corona. Such shapes are the result of zircon being exposed to various crustal conditions, from impact events to hydrothermal reactions (e.g., Corfu et al., 2003 and references therein). Contrary to the common opinion that zircon behaves extremely robust and inert, its behavior in various metamorphic conditions and during

deformation events can be highly variable (e.g., Reddy et al., 2007; Kohn et al., 2015; Kovaleva et al., 2015, 2016), even under low metamorphic grades (e.g., Dempster et al., 2008), especially in the presence of fluid. Mineral-fluid interaction can result in very "unusual" zircon textures.



In natural samples there are several known documentations of corona textures of zircon from igneous and meta-igneous rocks (e.g., Bingen et al., 2001; Söderlund et al., 2004; Austrheim et al., 2008), as well as from metapelites from Ivrea-Verbnano Zone (Pape et al., 2016). Such textures are found in rocks of different metamorphic grades, ranging from prehnite-pumpellyite to eclogite facies (Austrheim et al., 2008). It has been suggested that corona textures may evolve in magmatic

rocks as a result of slow cooling (Morisset et al., 2005), and in metamorphic rocks due to mineral-fluid reactions during progressive metamorphism (e.g., Bingen et al., 2001). It have been pointed out that the reactions with zircon precipitation in metamorphic rocks are often more efficient in the zones available for fluid, like fractures and shear zones (Bingen et al., 2001).

One of the first descriptions of zircon coronas in mafic meta-igneous rocks was done by Söderlund et al. (2004). The authors

attributed formation of secondary fine-grained zircon to the breakdown of baddeleyite in the presence of silica ("saccharoidal" zircon), and to consumption of minerals that have trace amount of Zr, such as ilmenite ("coronitic" zircon). In that case, both forms of secondary zircon precipitated under the increased metamorphic conditions. Bingen et al. (2001) have found hat-shaped zircon grains and coronas around ilmenite in granulites and amphibolites, which they considered to be a result of mineral reaction in fluid presence during metamorphism. Charlier et al. (2007) and Austrheim et al. (2008)

reported fine-grained zircon forming chains around ilmenite and rutile grains in meta-gabbros and suggested that zircon chains have grown around primary Fe-Ti oxides and, therefore, trace their former grain boundaries. Fine-grained zircon was reported to frame some rutile grains in the metapelitic septae from Ivrea-Verbano Zone (Ewing et al., 2013; Pape et al., 2016). Primary Fe-Ti oxides are considered to be either sources of Zr (e.g., Bingen et al., 2001; Söderlund et al., 2004; Pape at al. 2016), or, alternatively, having favorable surfaces as nucleation spots for zircon growth (Söderlund et al., 2004).

Tomkins et al. (2007) have experimentally produced zircon from Zr-rich rutile. As a result or rutile breakdown, zircon appears as thin exsolution lamellae or as euhedral individual small grains within rutile. The samples of metapelites (Ewing et al., 2013; Pape et al., 2016) from the Ivrea-Verbano Zone, could potentially serve as a natural example of such reaction, as these authors describe thin zircon needles in rutile and chains of fine zircon grains framing rutile.

## 1.2 Zr-bearing minerals and associated zircon precipitation

The abovementioned studies point to the conclusion that zircon dissolution and precipitation during metamorphism is not an independent process, but must be coupled with the growth of other minerals and with various mineral-fluid reactions in the host rock (e.g., Tomkins et al., 2007; Austrheim et al., 2008). During high temperature metamorphism, zircon may be partially or completely dissolved in the fluid (Ewing et al., 2014) and thus crystallize later, during exhumation and cooling of the host rock (Rasmussen, 2005; Kohn et al., 2015). Metamorphic precipitation of zircon, in some cases, could also be a

result of breakdown of various Zr-bearing phases (Davidson and van Breemen, 1988) such as garnet, amphibole, pyroxene and ilmenite (e.g., Fraser et al., 1997; Söderlund et al., 2004; Kelsey et al., 2008; Morisset and Scoates, 2008), hemo-ilmenite (Morisset et al., 2005), baddeleyite (Bingen et al., 2001; Söderlund et al., 2004), rutile (Tomkins et al., 2007; Morisset and Scoates, 2008; Kelsey and Powell, 2011; Ewing et al., 2013, 2014; Pape et al., 2016), epidote, clinopyroxene and titanite (Kohn et al., 2015). Zircon corona was even reported to grow from Martian baddeleyite as a result of shock metamorphism

(Moser et al., 2013). However, in mafic metamorphic rocks Zr is most often associated with Fe-Ti oxides (Bingen et al., 2001; Ewing et al., 2013, 2014), due to similar chemical properties with Ti.

Zr and Ti both belong to the group 4 in the periodic table, have close chemical properties and are usually regarded as relatively immobile trace elements (e.g., Mohamed and Hassanen, 1996). In the group of incompatible cations, Zr and Ti belong to high field strength (HFS) elements, which are smaller and are highly charged compared with large ion lithophile

(LIL) elements. The chemical similarities of Zr and Ti result in a positive correlation between them for most rock suites and



in their ability to replace each other in Fe-Ti oxides (e.g., Morisset et al., 2005). The fact that Zr oxides and Ti oxides are spatially related in many rocks also reflects their chemical similarities.

Thus, rutile and ilmenite are the main minerals that influence Zr mass balance in metabasites, in the absence of other Zr phases (e.g., Ferry and Watson, 2007; Tomkins et al., 2007; Morisset and Scoates, 2008; Ewing et al., 2013). It has been shown that in absence of zircon, rutile is also the main phase holding Zr and Hf in granulite-facies metapelites (Ewing et al., 2014). Zr is a common component of rutile, where its content can reach 10 000 ppm (e.g., Ewing et al., 2013); and thus Zr distributions often reflect the formation and decomposition of rutile, which may control the zircon stability (Kelsey and Powell, 2011). Zircon growth is frequently associated with the oxide transition from rutile to ilmenite during late-stage exhumation and cooling (Ewing et al., 2013). When rutile breaks down to form ilmenite or titanite, it expels Zr and thus causes the growth of zircon under a large variety of *P-T* conditions (Ewing et al., 2013). The temperature dependence of Zr solubility in other minerals (e.g., Zr-in-rutile) can have a fundamental impact on the zircon growth rate (Kohn et al., 2015), and is used for temperature estimations (e.g., Pape et al., 2016). Zirconium-in-rutile thermometer for rutile-quartz-zircon system has been calibrated by a number of authors (e.g., Watson et al., 2006; Ferry and Watson, 2007; Tomkins et al., 2007; Lucassen et al., 2010; Ewing et al., 2013), who have shown a large temperature- and pressure-dependent solubility of Zr in rutile.

Magmatic and metamorphic ilmenite could also be a reservoir of significant amounts of Zr (Bingen et al., 2001; Morisset et al., 2005; Charlier et al., 2007; Austrheim et al., 2008; Morisset and Scoates, 2008), which can reach more than 500 ppm (e.g., Morisset and Scoates, 2008). Bingen et al., (2001) describe coronas of zircon forming at the surface of granulite-facies ilmenite. One finding was also described for eclogites, where zircon coronas occur around rutile at a distance, indicating reaction of ilmenite replacement and marking the former grain boundary (Austrheim et al., 2008). Morisset and Scoates (2008) report 1-100 μm thick zircon coronas around ilmenite in mafic plutonic rocks. They consider it to be a result out-diffusion of Zr from ilmenite during slow cooling, as a result of precipitation form hydrothermal fluid.

In this study we report the occurrence of zircon coronas of various thickness (up to 1 μm thick and 5-20 μm thick) within Fe-Ti oxides in granulitic metapelites. This is the new textural relationships between zircon and host rutile, which has not been reported before. We show evidence supporting the idea that these structures are formed by mineral-fluid reactions, and helping to understand the initial mineral parageneses linked with evolutionary stages of the host rock. Presented textures reveal former reaction fronts. Such studies of precipitated zircon textures have potential implications for geochronology and may yield the isotopic age of the mineral-fluid reactions (Charlier et al., 2007; Ewing et al., 2013) and can provide temperature conditions for mineral reactions (Ewing et al., 2013; Pape et al., 2016). The latter possibilities, however, are not covered in this study and are the subject of future investigations.

## 2 Geological background and sample description

The sample material was collected in the Ivrea-Verbano Zone, Southern Alps (Val d'Ossola, Northern Italy). The Ivrea-Verbano Zone (IVZ) consists of a NE-SW trending, steeply dipping sequence of meta-sedimentary and meta-igneous basic rocks, ultrabasic mantle tectonites and a large underplated igneous complex (Fig. 1a). The sequence consists of predominant metasedimentary rocks in the SE and prevailing metabasic rocks and strongly depleted metapelites in the NW. Metamorphism increases progressively from amphibolite facies in the SE to granulite facies in the NW. The Ivrea-Verbano Zone is generally accepted as a section through the lower continental crust that experienced regional metamorphism during the uppermost Palaeozoic, tectonically overturned and uplifted. Today it is delimited by the Insubric line from the NW and by the Pogallo line from the SE (Brodie and Rutter, 1987; Barboza et al., 1999; Rutter et al., 2007; Quick et al., 2009).



Investigated samples came from the north-western part of the IVZ, were metapelites and metagabbro are completely (re-)equilibrated under granulite-facies conditions prevailing during crustal attenuation/extension and contemporaneous magmatic underplating between 315 Ma and 270 Ma (Rutter et al., 2007; Quick et al., 2009; Sinigoi et al., 2011; Klötzli et al., 2014). In the Val d'Ossola section, where the sampling took place, at Cuzzago peak *P-T* estimates are ~1.0 GPa and >

900 °C, based on the peak *P-T* estimates from the neighbouring Val Strona di Omegna (Redler et al., 2012). Other estimations (Sills, 1984) indicated granulite-facies *P–T* metamorphic conditions in the metapelites from the NW section of Val Strona at a maximum of 750±50 °C and 0.6±0.1 GPa. Zr-in-rutile temperatures of up to 850–930 °C were obtained for granulite-facies metapelites from Val Strona and Val d'Ossola (Luvizotto and Zack, 2009). Granulite-facies metamorphism has been accompanied by the anatectic melting of metapelites with the separation of restite material and removing of granitic

melt (Ewing et al., 2013).

The sampling locality near the village Cuzzago (Val d'Ossola) shows massive, non-foliated granulite-facies metasediments, known as stronalites. Stronalites locally reveal compositional layering which is formed during migmatization by alternating so-called "leucocratic injections" (Siegesmund et al., 2008) and restitic layers. This migmatization, or partial melting, took place at peak granulite-facies conditions (e.g., Ewing et al., 2013). The resulted compositional layering is moderately folded.

Stronalites are crosscut by a discordant layer of sillimanite-biotite-garnet gneiss (45°59'46.46" N/8°21'38.65" E, Fig. 1b). Similar rocks were interpreted to be a restitic material from a partial melting of metapelites (e.g., Barboza et al., 1999; Ewing et al., 2013; Pape et al., 2016). The restitic mineral assemblage in granulitic metapelites formed due to dehydration of biotite and decomposition of sillimanite and ilmenite, followed by the formation of garnet and K-feldspar, as well as rutile (e.g., Luvizotto and Zack, 2009; Pape et al., 2016). Studied sample was taken from this discordant layer (Fig. 1b), and represents a

weakly-foliated gneiss, broken by abundant faults that are normal to foliation. Foliation in the layer is striking NW (310°, angle 77°) and lineation is dipping to the NE (038°, dip angle 34°). No obvious kinematic indicators were observed in the host stronalites or in the sampled restitic gneiss. However, detailed structural investigations of the shear zones in the neighboring Val Strona revealed "a large number of macro- and microstructures" that provide consistent evidence of "sinistral shear operated over a long time span from the ductile to the brittle field" (Siegesmund et al., 2008 and references

therein). The authors emphasize that both brittle and ductile deformation may have operated simultaneously in granulitic metapelites during the formation of shear zones, which resulted in complex interactions of deformation microstructures.

Foliation of the sample is formed by a fabric of elongated garnet and sillimanite crystals 0.5-1 mm in length that compose 90 % of the sample. The stretching lineation is formed by elongated biotite crystals. The restitic mineralogy is composed of garnet, biotite and sillimanite with minor amounts of cordierite, ilmenite, rutile, K-feldspar and minor amounts of quartz.

Spaces between garnet and sillimanite grains and veins/fractures within the rock are filled by K-feldspar, quartz, ilmenite-rutile-quartz intergrowths and, sometimes, with a mixture of fine-grained minerals. The latter probably consists of various phyllosilicates: chlorite, muscovite/phengite (Fig. 1c-d, for the chemical composition see Table 1). Biotite is altered and contains numerous micrometer-size apatite needles and is mostly replaced by chlorite. Accessory minerals are zircon and monazite (Fig. 1d). When hosted by garnet, zircon forms roundish elongated crystals with aspect ratios from 1 to 3, and

lengths from 30 to 100 μm (Fig. 2a). When forming intergrowths with sillimanite zircon occurs as euhedral crystals with well-developed faces and forms triple junctions with the adjacent sillimanite grains (Fig. 2b). Sometimes, when hosted by fine-grained material that fills fractures, zircon crystals are fractured and fragmented. The fragments show irregular dissolved boundaries and finite lattice strain (Fig. 2c). Vermicular- and hat-shaped crystals and coronas formed by zircon usually are spatially associated with each other and attributed to the ilmenite-rutile-quartz or rutile-quartz intergrowth clusters (Figs. 3-

5). Therefore, it seems that the formation of the rutile-quartz aggregates is directly connected with zircon coronas formation. Ilmenite-rutile-quartz intergrowths fill transgranular fractures and pockets in the gneiss (e.g., Fig. 4a). These fractures are especially conspicuous at the thin section scale as light-brown veinlets that stretch mostly subvertically (Fig. 4a, arrows).



## 3 Sample preparation and analytical methods

Zircon textures were examined *in situ* using polished thin sections that were mechanically prepared with 0.25 μm diamond paste. Zircon grains were identified by backscattered-electron (BSE) imaging, and characterized by cathodoluminescence (CL) imaging for the internal growth features, using a FEI Inspect S scanning electron microscope equipped with a Gatan

MonoCL system (Faculty of Earth Sciences, Geography and Astronomy, University of Vienna, Austria). Imaging conditions were at 10 kV accelerating voltage, CL-image resolution of 1500*1500 to 2500*2500 pixels using a dwell time of 80.0-150.0 ms and probe current 4.5-5.0 nA. To identify the qualitative chemical composition of the host phases an EDX detector has been used. Orientation contrast images of zircon grains (Figs. 2a-c; 4f) were taken using a forescatter electron (FSE) detector on the chemically polished sample surface. FSE detector is mounted on the EBSD-tube of a FEI Quanta 3D FEG instrument

(Faculty of Earth Sciences, Geography and Astronomy, University of Vienna, Austria), which is equipped with a Schottky field emission electron source. Electron beam conditions were 15 kV accelerating voltage, 2.5-4 nA probe current using the analytic mode. Stage settings were at 70° tilt and 14-16 mm working distance. To investigate the host mineral phases, electron microprobe analyses were performed with the electron microprobe analyser (EMPA) Cameca SX 100 instrument equipped with 4 WDS spectrometers and an EDX system for high quality of quantitative chemical analyses (Faculty of Earth

Sciences, Geography and Astronomy, University of Vienna, Austria). Operating conditions were 15 kV accelerating voltage and 100 nA probe current. The detection limits in [ppm] for each EMPA analysis point are presented in Supplemental Table S1.

## 4 Results

### 4.1 Zircon microstructures and textures

Zircon textures, reported in this study, are coronas, by which we mean thin envelopes or shells (in 3D volume). Accordingly, in 2D plane of a sample they look like threads or worms (depending on the thickness and aspect ratio). Thus, in our description we will distinguish between "vermicular-shaped" and "corona" zircon aggregates, based on their thickness in 2D sections and on the aspect ratio of the aggregate. Specifically, vermicular-shaped zircon grains have thickness ≥5 μm, and aspect ratio 1:4 to 1:20; whereas corona zircon has a thickness ≤1 μm and aspect ratio more than 1:100 (e.g., Figs. 3-4).

Figure 3a shows a zircon aggregate composed of three large vermicular-shaped grains (indicated by "V", enlarged in Fig. 3b, c and e). These vermicular grains are 5 to 15 μm thick and 20 to 50 μm long and have diffuse or "auroral-light" (Corfu et al., 2003) CL zoning (Fig. 3b, c, e). Vermicular-shaped grains have curved (Fig. 3b) or ragged (Fig. 3e) boundaries, a crescent shape (Fig. 3c, e), and are often broken with multiple subvertical transgranular fractures (Fig. 3b, e). Some of these fractures can be traced in the host rock and they are filled with fine-grained phyllosilicates (Fig. 3e), which suggests that the

vermicular zircon formed earlier than the main cataclastic event and subsequent phyllosilicate growth. Furthermore, one of the fractures in vermicular grain e (figure 3e, highlighted by a circle) is partially healed by low-CL zircon material. This indicates that some precipitation of zircon has occurred after fracturing. Vermicular-shaped zircon grains are hosted by thin rutile-quartz intergrowths, where rutile forms <1 μm thin and 1-3 μm long needles (Fig. 3b, matrix). Such needle-shape evidences of rapid rutile recrystallization and re-equilibration during the metamorphic evolution.

Matrix around the vermicular grains c and e contains abundant and continuous thin zircon coronas and fine-grained zircon grains arranged in chains (coronas: highlighted by arrows, indicated by "C" in Fig. 3 c-e; chains: indicated by arrows with label "Ch" in Fig. 3d). Both zircon coronas and grain chains have distinguishable CL-response (Figs. 3c-e). Coronas occur as continuous threads that form splits (Fig. 3e, lower right), or isolated tangles (Fig. 3d, center). Zircon coronas form especially dense network between vermicular grains c and e (domain d, Fig. 3d), and they are often attached to the larger vermicular



zircon grains (Fig. 3c-e). Coronas are ≤ 1 μm thick, and distributed rather randomly inside the rutile-quartz intergrowths (Fig. 3c-e, arrows). Sometimes coronas extend outside of the rutile-quartz intergrowths, where they follow the phase boundaries between rutile-quartz aggregates and quartz. Some thin zircon coronas follow/are parallel to the fractures (Fig. 3a, highlighted by a circle), which supports the suggestion of zircon post-fracturing precipitation.

Another example of vermicular-shaped zircon aggregate associated with zircon coronas is presented in Fig. 4. This structure is found in a dark vein (Fig. 4a) that is filled with ilmenite-rutile-quartz intergrowths and quartz veins. The large zircon aggregate forms a W-shape and consists of two major fragments that are fractured (Fig. 4c, indicated by "V"). Its thickness varies from 5 to 20 μm, and the total length is about 200 μm. The W-shaped vermicular aggregate shows diffuse CL-zoning (Fig. 4d). At the lower right tip the aggregate is wedged. The CL image and the EDX map of Zr distribution show the blurred

trace around the lower right tip (Fig. 4d-e, pointed with the gray arrow). This indicates that the zircon aggregate continues deeper into the sample at a shallow angle, and the signal is still documented by CL and EDX from below the surface (few μm). As such the aggregate represents a shell-type corona in 3D. The W-shaped vermicular aggregate is surrounded by <1 μm thick coronas that are attached to it (Fig. 4b-e, pointed by arrows and indicated by "C"). Here coronas occur as continuous threads and are mostly distributed along the phase boundary between quartz and rutile-quartz intergrowths, or

between rutile and rutile-quartz intergrowths (Fig. 4c, bottom, coronas are indicated by arrows). Coronas show recognizable CL-response (Fig. 4d, arrows).

The vermicular W-shaped zircon grain in this aggregate is plastically-deformed at its tip, which is indicated by an orientation contrast image (Fig. 4f, black arrow). Rotation of the lattice on this tip reaches 7° with respect to the undeformed lattice (Kovaleva et al., 2016). This observation indicates that zircon vermicular grains preexisted not only brittle deformation of the

host rock, but also shearing and ductile deformation. It has been suggested by Siegesmund et al., (2008) that ductile and brittle deformation of the host rock could have occurred simultaneously during the formation of shear zones. Our observations support this idea.

Several more examples of coronas and vermicular-shaped grains are shown in Figure 5. In Fig. 5a (left) rutile-ilmenite intergrowths are filling the fracture inside the garnet next to a rectangular zircon grain. Zircon coronas are tracing the

boundary between garnet and rutile-ilmenite intergrowths and are connected to the rectangular zircon grain, so that the latter acquires a hat-like shape (Fig. 5a, marked as "*Hat*"). The hat-shaped grain has a flat surface adjacent to the rutile-ilmenite intergrowth. Such features were described earlier, for example, by Bingen et al., (2001). At the right hand side in the Fig. 5a, a rutile-phyllosilicate aggregate fills the pocket between sillimanite and garnet and contains vermicular-shaped fractured zircon grains (Fig. 5a, indicated as "V"). In Fig. 5b zircon coronas and chains of fine-grained zircon are framing the outer

boundary of rutile-ilmenite and rutile-quartz intergrowths (marked as "C"), and are associated with the larger zircon grains (marked as "*Zrn*"). In Fig. 5c rutile-quartz-phyllosilicates intergrowths fill the space between garnet and sillimanite grains. Zircon coronas are located inside the intergrowths and at the boundary between intergrowths and garnet, spatially associated with the larger zircon grain enclosed in sillimanite (marked as "*Zrn*") below the rutile-quartz aggregate. Locally zircon coronas follow fractures in garnet. In Figure 5d zircon coronas and chains are distributed inside the rutile-ilmenite-quartz

intergrowths and also outline the boundary between these intergrowths and a quartz grain. Coronas are elongate (up to 70 μm), have thickness up to 8 μm, which is actually closer in thickness to vermicular-shaped grains. However, the large aspect ratio and general appearance resemble coronas that are described above. Two larger roundish zircon grains (marked as "*Zrn*") are spatially associated with these coronas, and the largest one has outgrowths pointed towards the rutile-quartz aggregate.

The microstructural data show that the zircon "coronas" can be observed anywhere around and within rutile-quartz-

phyllosilicates, ilmenite-rutile and ilmenite-rutile-quartz aggregates (Figs. 3-5), and usually in the presence of the larger zircon grains. These larger zircon grains can be both vermicular-shaped aggregates and detrital grains. At the same time, not





all rutile-quartz intergrowths are associated with zircon coronas. In many cases only vermicular-shaped grains can be found with the rutile (e.g., Fig. 5a, right hand side), or no associated zircon is observed (Fig. 1c).

### 4.2 Microprobe data

Mineral electron microprobe data are presented in the Tables 1 and 2. $X_{Fe}$ of garnet is systematically lower in the cores, than
in the rims and in the fragments involved in the mineral reactions. The same applies to the biotite (Table 1). Garnet rims are also systematically enriched with Mn, compared to the cores. Chlorite is presented by two generations – (1) it forms pseudomorphoses after biotite. In this case former biotite crystals are full of micrometer-sized grains of apatite. In the second case (2) chlorite occurs in the matrix together with the fine-grained phyllosilicates and K-feldspar. Two chlorite generations are compositionally similar (Table 1). Compositions of Fe-Ti oxides are presented in the Table 2 and demonstrate that rutile
is much higher in $SiO_2$ content, than ilmenite. Rutile is also slightly enriched in such elements as Al, Cr, and Nb, but lower in Mn (Table 2), compared to ilmenite.

### 5 Discussion

#### 5.1 Mineral reactions

Mineral textures (Figs. 1c-d, 2c, 3a, 4b, 5) and microprobe analyses (Table 1) indicate that the initial granulite-facies
minerals such as garnet and sillimanite were intensively fragmented. The fragments have ragged edges and are plastically deformed, dissolved and altered (Figs. 1c-d; 2c; 4a; 5a-c). Primary ilmenite was partially/entirely decomposed to the secondary phases. The following features are regarded as evidence for intensive mineral-fluid reactions with possible hydration (e.g., Rajesh et al., 2013) of the dry restitic granulite-facies rock: reaction rims around fragments of granulite-facies minerals (e.g., Fig. 1c); fine-grained phyllosilicate mixture that fills fractures (e.g., Fig. 1c-d); K-feldspar (Fig. 4b) and quartz
veins (Fig. 4a). Garnet rims and smaller fragments, as well as biotite fragments, are enriched in Fe, which implies outward-diffusion of Mg and inward-diffusion of Fe from the environment. Ilmenite cores are surrounded by rutile rims (Figs. 1c, 5a-b, d), which implies decomposition of ilmenite with the formation of rutile that should result in the migration of Fe into the matrix, to form Fe-rich phyllosilicates and diffuse into the rims of garnet and biotite. Slow decomposition of Zr-rich ilmenite causes the growth of zircon exsolution lamellae with formation of thick vermicular-shaped grains (e.g., Figs. 3b, c, e; 4b-c).
Based on the mineral textures, we suggest the following mineral-fluid reaction that would involve elevated activity of the water fluid:

$$Grt_1 + Sill + Bt_1 + Ilm + H_2O_{Fl} = Grt_2 + Bt_2 + Chl\ (Phyl) + Ap + Kfs + Qtz + Rut_{Zr\text{-}rich} + Zrn_{verm} \qquad (1)$$

Where: $Grt_1$ = primary garnet, $Sill$ = sillimanite, $Bt_1$ = primary biotite, $Ilm$ = ilmenite, $H_2O_{Fl}$ = water in the fluid phase, $Grt_2$ = Fe-rich garnet, $Bt_2$ = Fe-rich biotite, $Chl$ = chlorite, $Phyl$ = other Mg-Fe phyllosilicates, $Ap$ = apatite, $Kfs$ = K-feldspar, $Rut_{Zr\text{-}rich}$ = Zr-rich rutile, $Qtz$ = quartz, $Zrn_{verm}$ = zircon vermicular grains. This reaction may unfold in several stages during the evolution of the rock (prograde to peak metamorphism, and the beginning of cooling).

Water-rich fluid could have been sourced from the decomposing biotite (e.g., Pape et al., 2016). Together with Fe from
ilmenite, and Mg diffusing out of garnet and biotite rims, fluid presence needs to compensate the formation of the large amount of Mg-Fe phyllosilicates in the matrix (Fig. 2c, Table 1). K from biotite and Al from sillimanite would be transported by water fluid and allow/favor the growth of K-feldspar rims (e.g., Fig. 4b). Excess of Ti, Hf and Zr from ilmenite would be responsible for the formation of rutile and zircon. Excess of Si from the fragmentation and dissolution of garnet and sillimanite would form quartz in the matrix, and react with Zr to form "vermicular" zircon. Newly formed rutile is enriched





in trace elements that could be a result of the decreased volume of Fe-Ti oxides after the reaction (1). It is also enriched in silica. The elevated content of silica may indicate the solution of $SiO_2$ in rutile (Taylor-Jones and Powell, 2015), which would play a role in a further reaction. Mn from ilmenite supposedly diffused into reacting garnet rims, some of which are slightly Mn-enriched (Table 1). Apatite-forming elements such as P, F, Cl and OH could be derived either from the de-composing

biotite, or were delivered by the water-rich fluid as components of a water brine from, for example, dissolution of monazite. However, occurrence of apatite needles inside biotite grains points to the strong genetic and chemical relationships between these two minerals.

Temperature estimations were done using garnet-biotite thermometer, which was applied to the microprobe data. Various calibrations of this geothermometer (Thompson, 1976; Holdaway and Lee, 1977; Ferry and Spear, 1978; Hodges and Spear,

1982; Perchuk & Lavrent'eva, 1983; Bhattacharya et al., 1992) gave the range of temperatures for the garnet-biotite cores of 570-700 °C; for the inner rims 800-860 °C; and for the outer rims 820-1090 °C. Estimations were done for pressures of 0.7 and 1.0 GPa, which did not have a significant effect on the temperatures. Thus the major mineral assemblage indicates prograde metamorphism with the maximum peak temperature of 1090 °C, which is in agreement with the previous estimations (Redler et al., 2012). It is, however, possible, that garnet and biotite rims were affected by diffusion from the host

environment.

Ewing et al., (2013) described partial replacement of rutile by other phases, characteristic for all granulitic metapelites from IVZ. In our sample we observe recrystallization of rutile with the formation of fine rutile-quartz intergrowths and zircon coronas around them. This has happened at the later stages of rock evolution, when the temperature decreases so that the Zr solubility in rutile also decreases. Therefore, we suggest the following reaction:

$$Rut_{Zr-Hf-rich} + SiO_2 = Rut_2 + Zrn_{coronas} + Qtz \qquad (2)$$

Where $Rut_{Zr-Hf-rich}$ = $SiO_2$-, Zr- and Hf-rich rutile resulted from decomposition of ilmenite, $Rut_2$ = $SiO_2$-, Zr- and Hf- depleted rutile that forms intergrowth with quartz $Qtz$ and zircon coronas $Zrn_{coronas}$. The source of $SiO_2$ could also be the same rutile

grains, which can contain comparatively large amounts of dissolved silica, which closure temperatures are high (Table 2; Taylor-Jones and Powell, 2015). During the dissolution-precipitation this silica was exsolved and formed thin intergrowths with the newly precipitated, trace elements-free rutile (e.g., Figs. 3, 4c) Zircon coronas are very thin that is the evidence of fast reaction rate and of large solubility of Zr in rutile (Tomkins et al., 2007; Ewing et al., 2013, 2014; Pape et al., 2016). Rutile-quartz intergrowths reveal some evidence of extreme deformation (non-equilibrium crystallization, fracturing). So the

reaction (2) might have been triggered by seismic deformation, which have been documented in metasediments of the Ivrea-Verbano Zone before (e.g., Pittarello et al., 2012; Kovaleva et al., 2015).

Reactions (1) and (2) have taken place under the elevated metamorphic temperature and in fluid presence that made the matrix minerals highly reactive. The preexisting mineral assemblage (garnet, sillimanite, biotite and ilmenite) indicates granulite-facies conditions, and the newly forming minerals (chlorite, phylloslicates) correspond rather to the greenschist

facies. Therefore, we suggest that mineral reaction (1) followed the peak of granulite-facies metamorphism and most probably occurred at the exhumation stage, but before the ductile and brittle deformation stage. The latter supported by observations of plastically-deformed vermicular zircon formed in the reaction (1), and the brittle fractures that crosscut the same mineral aggregates (e.g., Fig. 3e). On the other hand, formation of rutile in the granulites of the IVZ is often attributed to prograde and peak granulite metamorphism (e.g., Luvizotto and Zack, 2009; Ewing et al., 2013; Pape et al., 2016). Thus,

observed phyllosilicates in our sample might have formed at later stages of the rock evolution, later than decomposition of ilmenite (e.g., chloritization in the end of exhumation stage, fracturing and filling the fractures). In that case, reaction (1) should have covered a long time span – from the prograde and peak metamorphism to the exhumation stage.



In contrast with the thick coronas, formation of thin zircon corona during the reaction (2) occurred shortly before to simultaneous with fracturing, because we some of thin coronas are parallel to fractures or fill them (Fig. 3a; 5c). According to Luvizotto and Zack (2009), Zr-in-rutile temperatures are lower than the peak metamorphic temperatures, thus we suggest that the re-crystallization of rutile with exsolution of Zr (2) have happened after the peak metamorphism, at the cooling and

exhumation stage.

### 5.2 Zircon textures

Zircon grains hosted by garnet, sillimanite and fine-grained phyllosilicate matrix (Fig. 2) represent detrital grains overgrown by the main mineral phases during metamorphism. Zircon in garnet shows euhedral shapes and concentric growth zoning (Fig. 2a); zircon in sillimanite has detrital cores that are overgrown by a metamorphic rim (Fig. 2b). Metamorphic rims in

zircon form triple junctions and intergrowths with the host sillimanite (Figs. 2b; 5b-c) that indicates their simultaneous growth at the peak of granulite-facies metamorphism. After the peak metamorphic conditions these detrital zircon grains seem to be mostly inert and therefore, are well-preserved. Zircon grains hosted by the fine-grained phyllosilicate matrix are the most deformed ones – they are often fractured, and show slightly dissolved/corroded surfaces (Fig. 2c). These latter grains might have been involved in mineral reactions and were exposed to the peak and post-peak metamorphic fluids,

because are located in fractures. The dissolved material from their surfaces might be transported with a fluid and be a partial source for the new zircon generation (i.e. for "vermicular" and "corona" zircon aggregates), but alone it would not be enough to form large vermicular aggregates and extended hundreds-of-micrometers long zircon coronas (e.g., Fig. 4c). Thus we suggest that the other source(s) of zircon material (Zr) would be necessary (see reactions 1 and 2).

Zircon vermicular-shaped aggregates (or thick coronas) and thin coronas (Figs. 3-5) seem to be of a different origin than the

detrital grains (Fig. 2). Metamorphic origin of sub-micron sized zircon grains in metapelites has been demonstrated by Dempster et al., (2008), who argued that the metamorphic growth of zircon should be preceded by dissolution of detrital grains with the fluid. However, as it has been shown by other authors, zircon can grow from other Zr-bearing phases as a result of mineral reactions and as a mineral response to the changing conditions (Bingen et al., 2001; Söderlund et al., 2004; Austrheim et al., 2008; Ewing et al., 2013, 2014; Kohn et al., 2015; Pape et al., 2016). Metamorphic zircon in granulite-

facies rocks would not be the product of peak metamorphism, and grows during the retrograde evolution (Tomkins et al., 2007). Formation of zircon coronas could be interpreted as a result of expulsion of significant amounts of Zr from ilmenite and rutile during cooling after the granulite facies (Bingen et al., 2001; Ewing et al., 2013, 2014; Kohn et al., 2015). At lower temperatures rutile recrystallizes, and incorporates progressively less Zr (Ewing et al., 2013) than the high-temperature rutile, according to Zr-in-rutile thermometer models (Watson et al., 2006; Ferry and Watson, 2007). Thus, the access of Zr in the

cooling system should be hosted by other Zr-bearing phases, most often zircon and baddeleyite (e.g., Pape et al., 2016). Crystallization of thin ($\leq 1$ μm) zircon coronas indicates rapid cooling resulted in the recrystallization of Zr-bearing rutile, for example, by dissolution-reprecipitation, when the trace elements are being expelled from the host grain (Ewing et al., 2013).

Based on textural observations, we suggest that the new zircon growth in our sample occurred from Fe-Ti oxides in two stages. Mineral reactions (1) and (2) resulted in the formation of thick ("vermicular") and thin coronas accordingly:

(1)    Vermicular textures of zircon have not been described before, but they resemble zircon coronas with their shape (e.g., Bingen et al., 2001), being much thicker than what have been previously observed. Vermicular grains continue in 3D volume as the curved envelope-type surfaces (Fig. 4b-e). Therefore, we can interpret vermicular textures as evolved coronas. The thickness of these coronas should be controlled by reaction and cooling rate (Kohn et al., 2015). The slower the reaction and/or the slower the cooling is, the thicker should be the forming reaction

rims. We observe that rutile grows around ilmenite cores, therefore suggest that ilmenite is the primary Fe-Ti oxide in the granulite (Figs. 1c-d; 5a-b, d). Before and the peak granulite conditions ilmenite was the main host phase for



Zr, together with the primary detrital zircon (Bingen et al., 2001). At the peak grade and initial cooling stage ilmenite started to decompose to rutile (Ewing et al., 2013). The expelled Zr, which was not incorporated into the growing rutile, formed new zircon (Fig. 6). The reaction resulted in the volume decrease of the Fe-Ti oxides. This suggestion based on the observation showing that the newly grown rutile is enriched in trace elements comparing to ilmenite (Table 2). At the slow reaction rate the zircon coronas were growing thick, and formed exsolution lamellae intergrowths with the newly forming rutile (e.g., Fig. 4b-d). Formation of exsolution lamellae was described for many metamorphic minerals (e.g., Zhang and Liou, 2000). Thus, vermicular-shaped zircon that forms intergrowths with the rutile (e.g., Fig. 4c), probably represents ilmenite decomposition products.

Another process that could potentially have taken place at this stage is the dissolution of primary detrital zircon, where it was adjacent to ilmenite (Fig. 6). The dissolved material must have been precipitated within the forming rutile-ilmenite aggregate. This dissolved and precipitated small amount of zircon might have served as a seed for the newly formed vermicular-shaped zircon grains.

(2)      After the initial cooling, during further cooling at the exhumation stage (Ewing et al., 2014; Kohn et al., 2015), the solubility of Zr in rutile decreases and the zircon coronas grow rapidly around and inside the recrystallizing rutile grains. The thin needle-shaped 1-3 µm long rutile grains are evidence for rapid, non-equilibrium re-crystallization of rutile (e.g., Fig. 3b, d, e). Thin intergrowths of rutile with quartz suggests exsolution of previously dissolved SiO$_2$. This re-crystallization may be a result of co-seismic deformation, which might be also responsible for cataclastic nature of the studied rock and crystal-plastic deformation of mineral phases (see for example intense finite lattice strain in zircon and surrounding mineral fragments, Fig. 2c). The rutile grains that remain non-recrystallized usually occur in intimate contact with the rutile-quartz intergrowths and are separated from them by zircon coronas (Fig. 4c, arrows). At this stage the reaction rate is faster than during stage (1); and, as a result, the zircon coronas are thinner (Figs. 3-5). Nucleation of newly forming zircon starts around earlier zircon grains, similar to the low-temperature textures described in Rasmussen (2005). Zircon coronas in our sample are different from those described in Bingen et al. (2001), Charlier et al. (2007), and Morisset et al. (2008), where coronas are observed only at the boundary of the (former) ilmenite grains. They are also different from the coronas described in Austrheim et al., (2008) and Pape et al., (2016), where zircon forms continuous chains or closed contours of small grains. However, zircon coronas in all cases (described in earlier literature and here) represent shells around the reacting grains in 3D section (Bingen et al., 2001). Textures, indicating the reaction fronts of rutile recrystallization, have not been found by Pape et al., (2016) in their samples, even though they were looking for these features. In contrast, in our sample we clearly observe former reaction fronts formed by tangled and split zircon coronas within re-crystallized rutile aggregates. Tangled coronas are the evidence of different directions of the reaction fronts that met in one point (Fig. 3e; 4c; 5b; white arrows show the directions of the reaction front). The chains of small zircon grains are effectively the same as zircon coronas and are similar to those described in Austrheim et al., (2008). The hat-shaped zircon grains are formed by coronas that are connected to the larger zircon grains at the boundary with the rutile (Fig. 5a), and thus represent aggregates formed by different zircon generations. Flat face of a "hat" adjacent to the Fe-Ti oxide may indicate slight dissolution of zircon that occurred during stage (1) (see above).

Both mineral reactions occur earlier than the ductile deformation and brittle fracturing of the host rock. Thus we can attribute these reactions to the peak metamorphism, initial cooling (1) and early exhumation stage after the peak metamorphism (2), apparently, before/at the beginning of the formation of the shear zones (Siegesmund et al., 2008). Some minor precipitation and growth of zircon could have continued after fracturing (e.g., Fig. 3a, e, highlighted by circles). Extreme fracturing of the



rock (e.g., Fig. 4a) and crystal-plastic deformation of zircon (Figs. 2c; 4f) indicate the extreme conditions of post-peak metamorphism deformation (e.g., Siegesmund et al., 2008).

The thickness of the zircon coronas can be taken as evidence not only for the reaction rate, but also for the zircon growth rate that depends on Zr solubility in different minerals (Kohn et al., 2015). Solubility of Zr in ilmenite is much less than in rutile,
therefore the zircon coronas can grow thicker around Zr-bearing metasedimentary ilmenite in order to diffuse more Zr outwards. Expelled Zr reacts with the silicic matrix and forms zircon (Kohn et al., 2015).

Not all rutile aggregates in our sample are associated with zircon coronas. Similar case of diversity in appearance of rutile grains in one sample were described by Pape et al., (2016) for IVZ metapelites. This can be, for example, due to (a) thin section cut that does not reveal associated coronas; or (b) local re-crystallization of rutile (e.g., due to locally elevated strain,
inhomogeneous distribution of fluid, etc), so that the rest of rutile still contains significant amount of Zr. Possibility (b) leaves the field for Zr-in-rutile thermobarometry estimations (e.g., wing et al., 2013).

The schematic sketch showing the stages (1) and (2) of zircon coronas formation (Fig. 6) demonstrates that there was no significant net loss/gain of Zr from the system, and dissolved/exsolved Zr participated in the local environment. The possibility of two-stage exsolution of Zr from Fe-Ti oxides was suggested earlier by Ewing et al., (2013). Formation of
zircon coronas in our samples is restricted to metasediments, in contrast with the rocks described by previous authors (e.g., Bingen et al., 2001; Austrheim et al., 2008 and references therein). We did not observe zircon coronas in the host stronalites.

## 6 Conclusions and implications

In our study we demonstrate that zircon coronas can form within and around Fe-Ti oxides in metapelites, during and after peak granulite-facies metamorphism, before formation of the shear zones. We report a new textural relationship between
zircon and host rutile grains, as only exsolution needles of zircon in rutile and small zircon grains framing rutile were known for metapelites before (e.g., Ewing et al., 2013; Pape et al., 2016). Zircon corona aggregates form as a product of hydration mineral reaction conjugated with dissolution and element diffusion, followed by ductile and brittle deformation. Formation of zircon coronas occurred in two distinct stages, resulted in (1) thick (5-20 µm) vermicular-shaped grains formed presumably during slow decomposition of ilmenite to Zr-rich rutile; and in (2) thin (≤1 µm) corona aggregates and submicron-grain
chains formed due to rapid re-crystallization of Zr-rich rutile via dissolution-precipitation (Fig. 6). Mineral reactions with the Zr exsolution and zircon growth were fluid-induced and occurred at the sites of fluid access along veins and fractures. Two zircon-formation stages were separated in time and represent two evolution stages of the sampled rock, and could be therefore connected with the evolution of the Ivrea-Verbano Zone at a larger scale.

The detail study of the zircon corona textures can have a significant input in the trace element balance calculations for the
bulk rock, provide a tool for reconstruction of metamorphic and metasomatic mineral-fluid reactions, and help to derive the direction of reaction fronts. Moreover, such structures can be potentially used in geochronology for *in situ* dating of metamorphic evolution stages (providing that the appropriate high-resolution analytical technique is applied). The trace elements cam be measured to fingerprint different fluid infiltration/recrystalisation events. They can be used in thermobarometry for estimating the *P-T* conditions of the formation of non-recrystallized Zr-rich rutile, and the *P-T*
conditions of the Zr exsolution reaction. For the latter Zr-in rutile, Ti-in zircon and Si-in rutile thermometers can be applied.

## Author contribution



E. Kovaleva and U. Klötzli were responsible for sampling. E. Kovaleva performed laboratory work, SEM and EMPA analysis, data reduction and analysis, and drafted the manuscript. H. Austrheim and U. Klötzli conceptualized the study, oversaw the progression of the work and advised on interpretation.

## Acknowledgments

This study was funded by the University of Vienna (doctoral school "DOGMA", project IK 052). The authors acknowledge access to the Laboratory for scanning electron microscopy and focused ion beam applications, Faculty of Earth Sciences, Geography and Astronomy at the University of Vienna (Austria), and specifically Gerlinde Habler, who acquired orientation contrast images presented in this study. The authors are grateful to Rainer Abart, Claudia Beybel, Franz Biedermann, Sigrid Hrabe, Hugh Rice, and all colleagues of the FOR741 research group for fruitful discussions; to the Geologische

Bundesanstalt (GBA) of Austria and Christian Auer for access to the their SEM; to the Department of Geology in University of the Free State for support in writing this manuscript. Comments of two anonymous reviewers helped to improve the text greatly.

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



| Mineral | SiO$_2$ | TiO$_2$ | Al$_2$O$_3$ | MgO | CaO | MnO | FeO | BaO | Na$_2$O | K$_2$O |
|---|---|---|---|---|---|---|---|---|---|---|
| Grt core | 38.90 | 0.02 | 22.22 | 9.33 | 1.51 | 0.31 | 28.60 | n.d. | n.d. | n.d. |
| Grt core | 38.14 | 0.01 | 21.69 | 6.19 | 1.46 | 0.43 | 32.92 | n.d. | n.d. | n.d. |
| Grt core | 38.16 | 0.01 | 21.70 | 6.37 | 1.46 | 0.49 | 32.51 | n.d. | 0.01 | n.d. |
| Grt core | 39.01 | n.d. | 22.15 | 9.19 | 1.51 | 0.36 | 28.52 | n.d. | 0.02 | n.d. |
| Grt core | 39.10 | 0.02 | 22.09 | 9.26 | 1.50 | 0.32 | 28.80 | n.d. | 0.02 | n.d. |
| Grt core | 38.92 | 0.03 | 22.11 | 9.13 | 1.46 | 0.30 | 28.84 | n.d. | 0.03 | n.d. |
| Grt core | 38.83 | 0.01 | 22.00 | 8.34 | 1.41 | 0.35 | 29.92 | n.d. | 0.03 | n.d. |
| Grt core | 37.80 | 0.02 | 22.03 | 6.75 | 1.49 | 0.55 | 31.69 | n.d. | 0.01 | n.d. |
| Grt rim | 38.81 | 0.01 | 22.12 | 8.58 | 1.54 | 0.34 | 29.44 | n.d. | 0.01 | n.d. |
| Grt rim | 38.81 | 0.03 | 22.06 | 8.28 | 1.53 | 0.34 | 30.31 | n.d. | 0.03 | n.d. |
| Grt rim | 38.95 | 0.01 | 21.82 | 7.73 | 1.52 | 0.40 | 30.77 | n.d. | n.d. | n.d. |
| Grt rim | 38.35 | 0.01 | 21.95 | 7.77 | 1.51 | 0.42 | 30.35 | n.d. | 0.02 | n.d. |
| Grt rim | 38.61 | 0.01 | 21.59 | 5.67 | 1.46 | 0.52 | 33.25 | n.d. | 0.01 | n.d. |
| Grt rim | 38.39 | 0.07 | 21.78 | 5.96 | 1.46 | 0.53 | 32.82 | n.d. | n.d. | n.d. |
| Grt rim | 37.93 | 0.21 | 21.49 | 5.65 | 1.52 | 0.55 | 32.93 | n.d. | 0.02 | n.d. |
| Grt rim | 38.30 | 0.07 | 21.86 | 6.82 | 1.45 | 0.43 | 31.69 | n.d. | n.d. | n.d. |
| Grt rim | 38.34 | 0.04 | 21.75 | 7.15 | 1.48 | 0.43 | 31.37 | n.d. | 0.01 | n.d. |
| Grt rim | 37.35 | 0.01 | 21.12 | 3.85 | 1.50 | 0.87 | 35.51 | n.d. | n.d. | n.d. |
| Grt rim | 37.38 | 0.03 | 21.12 | 3.86 | 1.45 | 0.81 | 35.22 | n.d. | n.d. | n.d. |
| Grt rim | 37.65 | n.d. | 21.30 | 3.98 | 1.43 | 0.78 | 35.34 | n.d. | n.d. | n.d. |
| Bt core | 37.15 | 6.27 | 15.51 | 15.79 | 0.03 | 0.03 | 10.75 | n.d. | 0.11 | 9.83 |
| Bt core | 37.15 | 1.32 | 16.53 | 18.79 | 0.13 | 0.03 | 11.36 | n.d. | 0.10 | 9.16 |
| Bt rim | 34.81 | 1.74 | 17.17 | 14.41 | 0.04 | n.d. | 17.50 | n.d. | 0.12 | 7.83 |
| Bt rim | 35.22 | 2.82 | 16.59 | 13.55 | 0.02 | 0.04 | 18.27 | n.d. | 0.10 | 8.15 |
| Bt rim | 35.61 | 1.46 | 16.14 | 14.38 | 0.16 | 0.02 | 17.94 | n.d. | 0.09 | 7.38 |
| Bt rim | 34.75 | 1.27 | 17.16 | 13.37 | 0.03 | 0.04 | 19.60 | n.d. | 0.08 | 7.27 |
| Chl over Bt | 29.75 | 0.03 | 18.37 | 16.40 | 0.10 | 0.05 | 22.91 | n.d. | n.d. | 0.14 |
| Chl over Bt | 27.72 | 0.01 | 19.47 | 15.12 | 0.08 | 0.06 | 25.34 | n.d. | n.d. | 0.05 |
| Chl over Bt | 27.29 | 0.21 | 19.76 | 15.84 | 0.03 | 0.07 | 24.21 | n.d. | n.d. | 0.03 |
| Chl over Bt | 27.64 | 0.28 | 20.07 | 16.34 | 0.04 | 0.04 | 23.41 | 0.04 | n.d. | 0.04 |
| Chl over Bt | 28.09 | 0.40 | 19.07 | 15.97 | 0.06 | 0.06 | 24.07 | 0.04 | n.d. | 0.20 |
| Chl new | 27.14 | 0.02 | 21.48 | 14.10 | 0.03 | 0.09 | 24.90 | n.d. | n.d. | 0.07 |
| Chl new | 24.84 | 0.03 | 22.94 | 12.98 | 0.05 | 0.06 | 26.44 | n.d. | n.d. | 0.06 |
| Chl new | 28.47 | 0.02 | 18.16 | 15.53 | 0.06 | 0.05 | 24.92 | n.d. | 0.01 | 0.06 |
| Chl new | 29.00 | 0.08 | 18.92 | 15.73 | 0.09 | 0.06 | 24.07 | n.d. | 0.02 | 0.21 |
| Chl new | 29.44 | 0.06 | 19.27 | 15.65 | 0.07 | 0.06 | 24.44 | n.d. | 0.02 | 0.26 |
| Chl new | 28.70 | 0.02 | 18.38 | 15.42 | 0.08 | 0.07 | 25.45 | 0.05 | 0.04 | 0.09 |
| Chl new | 29.30 | 0.04 | 17.69 | 15.27 | 0.08 | 0.06 | 25.39 | 0.02 | 0.01 | 0.10 |
| Chl new | 31.95 | 4.44 | 17.50 | 11.51 | 0.04 | 0.05 | 21.81 | 0.04 | 0.04 | 4.64 |
| Phyl matrix | 44.34 | n.d. | 31.75 | 3.84 | 0.13 | n.d. | 6.06 | 0.28 | 0.33 | 7.70 |
| Phyl matrix | 49.04 | 0.01 | 32.24 | 1.68 | 0.04 | n.d. | 1.63 | 0.17 | 0.18 | 10.58 |
| Phyl matrix | 45.92 | 0.01 | 32.36 | 2.78 | 0.12 | n.d. | 4.12 | 0.47 | 0.32 | 8.63 |
| Phyl matrix | 43.30 | 0.18 | 32.40 | 4.12 | 0.15 | 0.05 | 7.16 | n.d. | 0.33 | 7.12 |
| Phyl matrix | 47.74 | 0.12 | 33.88 | 1.98 | 0.17 | 0.01 | 3.05 | n.d. | 0.42 | 8.67 |
| Phyl matrix | 44.89 | 0.21 | 32.84 | 3.28 | 0.10 | n.d. | 4.87 | n.d. | 0.37 | 8.39 |
| Phyl matrix | 44.91 | 0.02 | 32.38 | 3.38 | 0.17 | 0.02 | 5.17 | n.d. | 0.38 | 7.81 |
| Phyl matrix | 38.78 | 0.05 | 28.79 | 6.94 | 0.10 | 0.02 | 12.02 | 0.19 | 0.27 | 4.91 |
| Phyl matrix | 45.22 | 0.03 | 32.53 | 3.29 | 0.09 | 0.03 | 4.49 | 0.73 | 0.37 | 8.28 |
| Pheng/mus | 47.45 | 0.07 | 35.68 | 1.31 | 0.09 | n.d. | 1.65 | 0.32 | 0.40 | 9.53 |
| Pheng/mus | 48.41 | 0.17 | 35.13 | 1.34 | 0.14 | n.d. | 1.46 | 0.37 | 0.43 | 9.19 |
| Pheng/mus | 46.62 | 0.12 | 35.34 | 1.43 | 0.10 | 0.01 | 1.94 | 0.36 | 0.45 | 9.07 |

Table 1. Microprobe analyses results of the silicates compositions in the sampled rock, n.d. = not detected. See discussion in the text.



| Mineral | Ta$_2$O$_5$ | SiO$_2$ | TiO$_2$ | Al$_2$O$_3$ | Cr$_2$O$_3$ | Nb$_2$O$_3$ | MgO | MnO | FeO | NiO | Total |
|---|---|---|---|---|---|---|---|---|---|---|---|
| Ilm | 0.04 | 0.03 | 52.82 | 0.02 | n.d. | 0.11 | 0.05 | 0.85 | 45.18 | n.d. | 99.15 |
| Ilm | 0.03 | 0.14 | 53.07 | 0.12 | 0.02 | 0.12 | 0.07 | 0.85 | 44.30 | 0.02 | 98.73 |
| Ilm | n.d. | 0.15 | 53.72 | 0.12 | n.d. | 0.04 | 0.03 | 0.89 | 43.56 | n.d. | 98.52 |
| Rut | n.d. | 0.60 | 98.31 | 0.22 | 0.08 | 0.25 | 0.04 | n.d. | 0.44 | n.d. | 99.92 |
| Rut | n.d. | 0.80 | 98.46 | 0.26 | 0.08 | 0.40 | 0.01 | n.d. | 0.40 | n.d. | 100.42 |
| Rut | n.d. | 2.10 | 93.19 | 1.42 | 0.06 | 0.22 | 0.60 | n.d. | 1.46 | n.d. | 99.03 |

Table 2. Microprobe analyses results for the Fe-Ti oxides compositions in the sample, n.d. = not detected. See discussion in the text.



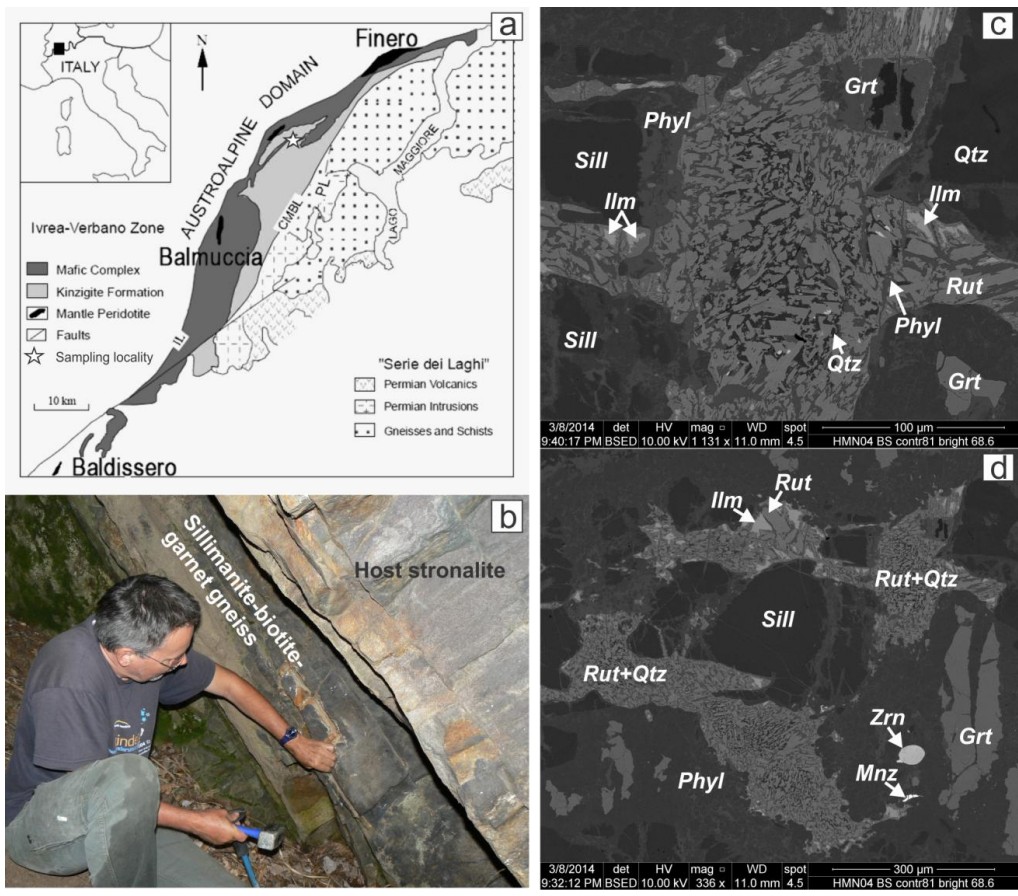

**Figure 1: (a)** Geological map of the Ivrea-Verbano Zone after Zanetti et al., (1999) with the sampling location indicated with the star. **(b)** Field photograph of the sampled outcrop with the discordant dyke-shaped body of the sillimanite-biotite-garnet, interpreted as restitic gneiss, hosted by mylonitized stronalite. **(c)-(d)** BSE thin section images with mineral parageneses. Note in (c) that the rutile aggregate contains ilmenite cores (bright gray), and forms intergrowths with two different phases: phyllosilicates from the reaction rim (gray shade, slightly darker than rutile) and quartz (the darkest phase); all indicated by arrows. Sill = sillimanite, Grt = garnet, Qtz = quartz, Ilm = ilmenite, Rut = rutile, Rut+Qtz = rutile-quartz intergrowths, Phyl = mixture of phyllosilicates, Zrn = zircon detrital grain, Mnz = monazite.





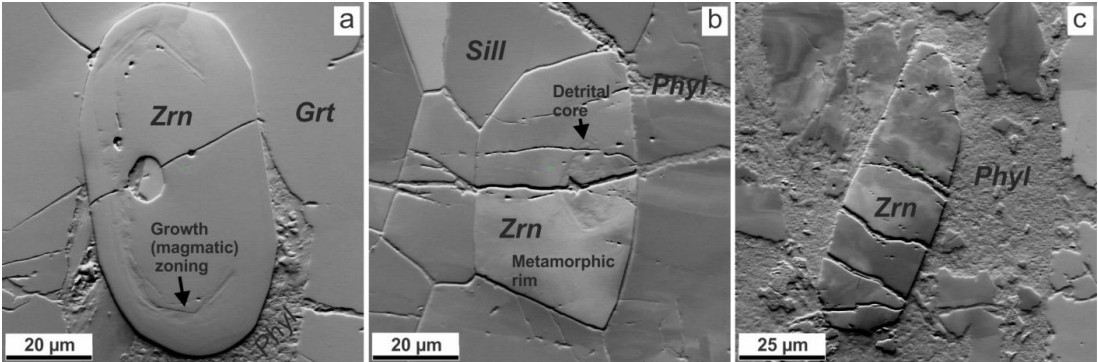

**Figure 2: Orientation contrast images of detrital zircon in the sampled gneiss: (a) zircon grain hosted by garnet, note the concentric growth zoning. (b) Zircon grain hosted by sillimanite, note the small detrital core (right hand side) and wide metamorphic rim. (c) Zircon grain hosted by a fine-grained matrix that fills the veins, note intensive change of orientation contrast, especially conspicuous in the upper part of the grain. Orientation contrast image indicates the presence of crystal-plastic deformation of the zircon grain and surrounding mineral fragments. Fracture surfaces appear to be dissolved. Mineral abbreviations as in Fig. 1c-d.**

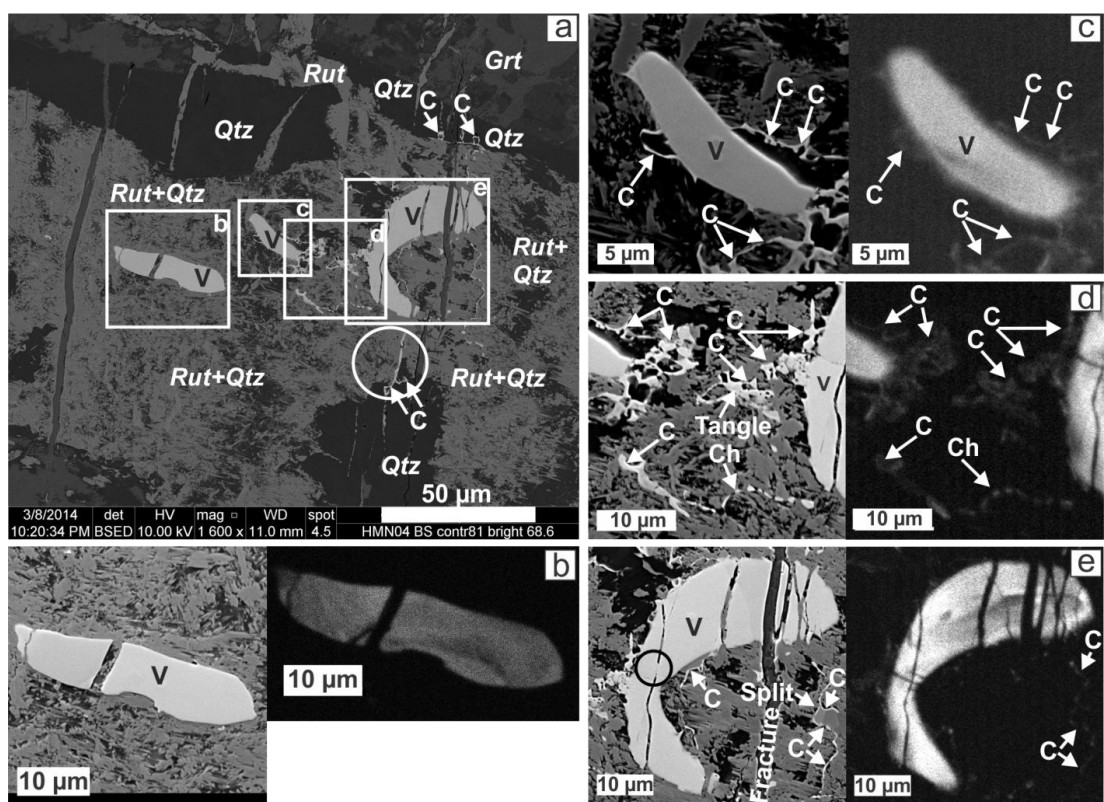

**Figure 3: (a) BSE image of the vermicular-shaped zircon aggregates ("V") and zircon chains ("Ch") coronas ("C"). Arrows indicate zircon coronas that trace quartz-rutile boundary or fill the cavities in quartz; circle highlights a corona that is parallel to the fracture. Mineral abbreviations as in Figure 1c-d. (c) - (e) Enlarged BSE (left) and CL (right) images of the areas indicated in (a). "V" highlights the vermicular-shaped zircon grains, "Ch" in (d) points to the chain of submicron-sized zircon grains, and "Tangle" points to the tangled occurrence of coronas. Arrows in (e) indicate the directions of the reaction fronts, and "Split" pints to the branching of zircon coronas. Circle in (e) highlights partially healed fracture in vermicular zircon.**

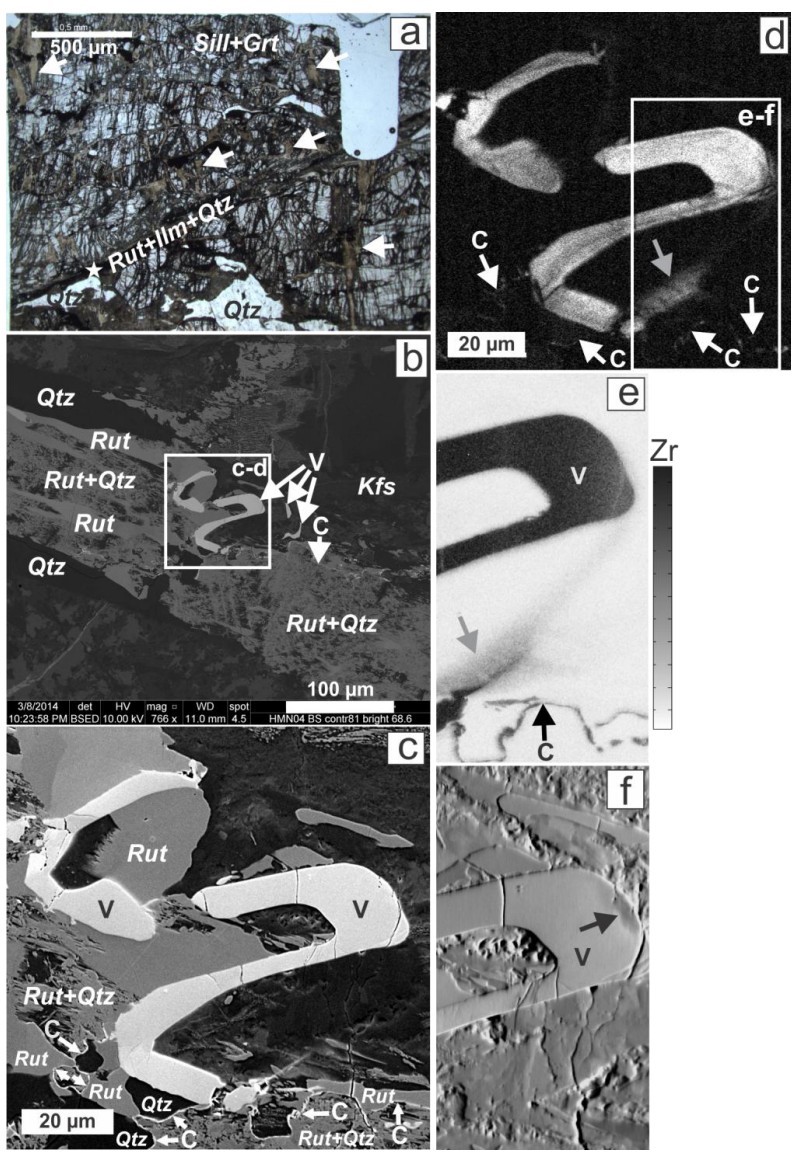

**Figure 4: (a) Plain-polarised light optical micro-photograph of a thin section with the position of the zircon aggregate (white star) shown in (b), note its position in the dark rutile-quartz vein that might have acted as a fluid channel. Arrows indicate intense fractures. (b) BSE image of the zircon aggregate intergrowth with rutile, rutile-quartz intergrowths and quartz. Mineral abbreviations as in Figure 1c-d, Kfs = K-feldspar. (c) BSE image of the enlarged area indicated in (b). Arrows point out the direction of reaction front, mineral abbreviations as in Figure 1c-d. (d) CL image of the enlarged area indicated in (b). White arrows as in (c), gray arrow points to the wedging zircon aggregate that continues below the surface. (e) Qualitative EDX intensity map for Zr of the area indicated in (d). Black arrow points out the direction of reaction front, gray arrow as in (d). (f) Orientation contrast image of the area indicated in (d). Plastically-deformed tip of vermicular zircon grain is pointed by arrow. "V" in (b)-(f) highlights vermicular zircon grains, "C" points to zircon coronas.**




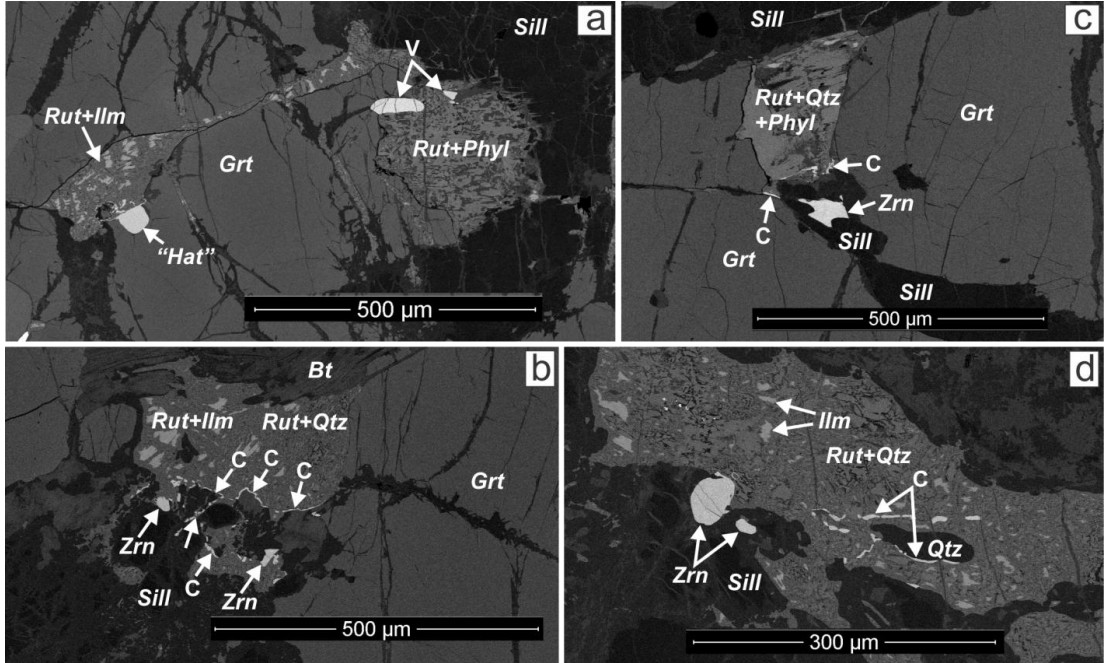

**Figure 5: BSE images of mineral reactions that contain zircon grains and associated zircon coronas. Mineral abbreviations as in Fig. 1c-d, Bt = biotite, Hat = hat-shaped zircon aggregate. "V" highlights the vermicular zircon grains, "C" points to the zircon coronas, Zrn = pre-existing zircon detrital grains, which have normal for this mineral shape, or from intergrowths with sillimanite like in (c).** 5 **Arrows in (b) show the direction of mineral reaction front.**

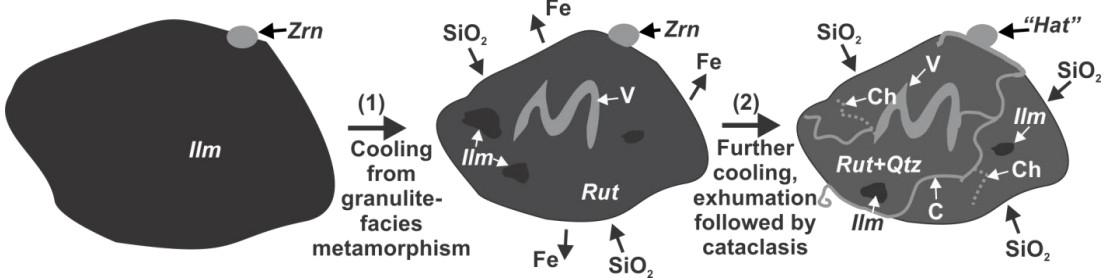

**Figure 6: Schematic sketch of the formation stages of zircon coronas. (1) Pre-existing metamorphic Zr-bearing ilmenite starts decomposing to rutile and forms vermicular-shaped zircon aggregates or exsolution lamellae as a side product. The system loses Fe** 10 **and requires SiO2 from the surrounding phases, the volume of Fe-Ti oxide decreases. Round metamorphic zircon experiences slight dissolution where it's adjacent to ilmenite, which results in the final "hat" shape. This happens during and soon after granulite-facies peak metamorphism at the initial cooling stage. (2) Further cooling and exhumation results in re-crystallization of Zr-rich rutile with exsolution of thin zircon coronas, followed by cataclasis, crystal-plastic deformation and formation of rutile-quartz intergrowths. System requires SiO2 from the environment. Abbreviations as in Figs. 1, 3 and 5.**