# Peer review of "Interpretation of zircon corona textures from metapelitic granulites of Ivrea-Verbano Zone, Northern Italy: Two-stage decomposition of Fe-Ti oxides"

_Solid Earth, 2016_

## Referee Comment (RC1) · N. Kelly (Referee) · 20 Jan 2017

Overview

The paper authored by Kovaleva and others presents a detailed description of metamorphic zircon textures in metapelitic granulites from the Ivrea-Verbano Zone, northern Italy. The authors suggest, on the basis of textural relationships, but limited geochemical data, that the zircon formed as a result of hydration reactions, where Zr was liberated from ilmenite and later from Zr-rich rutile during prograde metamorphism and then cooling after peak granulite facies metamorphism.

[Figure]

The textures presented are an intriguing record of zircon formation during solid-state metamorphic reactions, and this work represents a valuable contribution to our understanding of zircon growth mechanisms. However, the current structure of the paper and how observations have been presented make it difficult to properly evaluate the veracity of the interpretations made and conclusions drawn. Direct criticisms and suggestions for improving the presentation of the paper are made below, with further comments made on an annotated PDF version of the manuscript.

General Comments

* Introduction

Much has been published on the mechanisms of zircon formation during metamorphism, a brief presentation of which would be very valuable in this contribution. The paper does cover some of the basics of "corona textures" and reports of these in meta-morphic rocks, in particular meta-basic rocks. Although the focus on metabasites is curious, given this paper is on metapelites. However, it would be useful to see a recap of what we known about metamorphic zircon growth in all rock types as this does inform about the current study. In particular, this should cover differences between growth during major/accessory mineral breakdown, exsolution from accessory minerals, and precipitation from a fluid (vs growth as a result of major mineral decomposition during a retrograde hydration event). Some of this could be introduced during the discussion, but this background places the textures reported here in a clearer context. In this fashion, sections 1.1 and 1.2 could basically be integrated.

* Geologic background & sample description

The description of the outcrop and interpretations of the origin of the rocks needs a clearer description. I would suggest a layout similar to:

1) mesoscopic outcrop features 2) mineralogy of the rocks 3) pertinent textures 4) interpretation of metamorphic features (PT-conditions, PT-path) including integration

with published info.

A more detailed description of the sample taken would come later (see comments below).

Some other points - the stronalite (an outdated term that could do with replacing) has been interpreted by others to have formed through partial melting (not just "dehydration" during metamorphism). The leucosome in that rock may not just be injection of melt but melt formed in the rock itself. But what I find most curious is how a layer of metasediment can be discordant to layering in the host stronalite. Can this be explained better? But the focus here should really be to say that the rock studied has likely lost partial melt, so is somewhat restitic. And then later it is cut by brittle faults. Somewhere in there ductile deformation affected the sample, but from the outcrop description is this not by any means given context.

The mineralogy and textural description of the sample itself (page 4, line 34 onward) should be moved and inserted into the results section (4.1), as this first introduces the zircon occurrences and should be integrated with the rest of the textural descriptions. When the sample description (separate from outcrop description) is moved into section 4.1, there needs to be a clear and succinct description of the sample's major mineral textures. It is difficult to understand the relationships between prograde mineral textures (if present), peak mineral relationships, post-peak and hydration reaction textures (including the veins). From the description, I can't tell if there is more than one vein type – one that is dominated by Fe-Ti oxide minerals, the other dominated by fine-grained phyllosilicates, or are all a mixture of these? And the relationship of veins to mesoscopic and microscopic structures needs to be clearer – how can one deduce that a veins in thin-section "stretch mostly subvertically"? Is this an inference made from an oriented thin section, or just that when you look at the thin section it runs up and down?

* Zircon textures (Results)

This needs to be more systematic. At present, zircon textures are presented as different "occurrences" that have much overlap. This reads like they came from different veins or thin sections and so were considered separately. I see an advantage to integrating the descriptions into "types" of textures.

For example:

" Zircon occurs in 2 textural contexts (as 2 textural types). The first, . . . The second, . . . "

I see these types as the "vermicular" zircon (the coarser-grained zircon) and then the fine-grained "coronae" that occur either intergrown with Fe-Ti oxides mineral or in fractures. These may occur with different "vein types" but these are the basic textural types.

* Microprobe data

It is difficult to see, in it's current form, what this contributes to the overall understanding of the problem. Very little is made of the data (apart from showing some compositional zoning in garnet). Nothing is made of the analysis of the phyllosilicates. But perhaps most striking, is that zirconium hasn't been analyzed in rutile (or ilmenite). This would be very illuminating! This would lend credence to the interpretation that Zr was probably sourced from ilmenite and rutile, and could also be used to estimate the temperatures of formation. Zr-in-Rt is very doable my electron microprobe. It would be beneficial to have Ti-in-zircon, although I understand with the fine-grained nature of the textures this would be impractical.

* Mineral reactions

The cool thing about textures such as those presented here, is that they do provide insights into the reactions that could have led to growth of zircon during metamorphism. However, this is weakly dealt with in this paper.

First – reaction 1 makes no sense.

The metapelitic rocks have been shown to have experienced partial melting. Given the assemblage, I would suggest the reaction:

Bt + Sil + Qtz = Grt + Kfs + Qtz + Melt

This is not a "hydration" reaction, but a dehydration reaction, as water goes into the melt. This is also a prograde to peak reaction.

Formation of the veins, which host biotite, and the development of elevated Fe and Mn in garnet rims, likely occurred during retrograde hydration. Resorption of garnet during retrograde breakdown has been commonly reported as a mechanism to elevate Fe and Mn in rims. But, this mechanism MUST post-date the peak of metamorphism. While one can accept that the rock could have been hydrated from an external source and so could theoretically form during prograde metamorphism, the description of the mineralogy and textures does not support this. If you disagree with this interpretation, there must be an argument presented for it, and it is not present in the manuscript.

There also needs to be a more developed discussion of the mineral relationships in the veins. For example, rutile is interpreted to have formed from the breakdown of ilmenite, which still occurs as relics in the veins, which (at least some of) are dominated by rutile-quartz intergrowths. What was the mineralogical make-up of the veins before reaction? Does ilmenite (or rutile for that matter) occur elsewhere in the rock, or just confined to veins? Can you mass balance any reaction (even with a thought to open system)? Did Ti come in with H2O and SiO2? What is the distribution of new zircon relative to the different vein types – is there less zircon when less rutile (and ilmenite)?

* Zircon textures

Leading on from the above discussion, are questions about whether fine-grained zircon formed from solid-state mineral breakdown (rutile, ilmenite) or due to exsolution of Zr from these minerals. The assumption in the paper is that this is just mineral breakdown-growth. However, many papers have suggested fine-grained textures represent exsolution during cooling.

Without a clear understanding of the veins themselves, I can only speculate, but it seems to me that a viable option for the formation of these textures involves the coarser zircon grains forming at high-T. They could have formed in the presence of crystallizing melt (e.g., zircon in Fig. 5 is seen associated with Sil, or included in garnet). With cooling, Zr was less compatitible in rutile, so began to exsolve, hence the close textural relationship with rutile, and the lower-T not allowing grain coarsening. No doubt presence of fluids aided Zr mobility – hence presence along fractures and major grain boundaries with coarser-grained garnet or quartz. But evidence for pure metasomatic-driven reactions is not demonstrated here.

Some comments on figures:

* there are some wonderful textures here. However, they are a little too small to see in some cases. Consider re-arranging and increasing the size so the relationships are clearly visible.

* be sure to define all abbreviations – what is Phyl?

* there needs to be some clearer photomicrographs that show the metamorphic textures and the relationship to veins. If you want to push the idea of fluid-driven reaction, it would also help to present a systematic description (in the images) of the veins: the hydrous veins, rt-qtz intergrowths, etc. . . The current images make this very difficult to evaluate.

* while schematic figures are excellent to explain the processes invoked, it would help if they properly described the textural development. In figure 6, the process starts with a single grain of ilmenite, which reacts to form a single grain of rutile, which reacts to for rutile-quartz aggregates. I don't recall this being described in the text. And in any case, the paper presents these textures as occurring in veins. I am left confused about the process here.

Please also note the supplement to this comment:
http://www.solid-earth-discuss.net/se-2016-164/se-2016-164-RC1-supplement.pdf
* * *

---

## Referee Comment (RC2) · F. Corfu (Referee) · 4 Feb 2017

The paper presents an interesting case of secondary zircon growth as a result of mineral reactions which liberate Zr during cooling and exhumation from high grade conditions. Although the growth of zircon as a consequence of the break-down of ilmenite and rutile, or diffusion of Zr from these minerals during cooling has been described before, the present paper adds an important dimension documenting zircon formation during retrogression even at greenschist facies conditions.

Although one can follow the descriptions and discussions reasonably well, some parts

are rather confused and there is much repetition. Chapter 1 introduces the subjects and reviews the previous literature, but then much of this is repeated again in the second chapter, and then again in the discussion. The latter needs a thorough restructuring, ideally discussing the changes in mineralogy, structures and zircon features and evolution in a logical time progression. Here the discussion starts with the late events eventually getting to the early stages, and circling around and back several times. Chapter 5.2 is mainly a lengthy repetition of what has been said before, with a number of contradictory statements added in.

I have marked the file and added some questions and comments there.

My suggestion is to do a serious restructuring and condensation of the paper, sharpening the logic and cutting out the repetitions. A slender paper about 1/3 in size should be a result which the readers will greatly appreciate.

Febr. 4, 2017 F. Corfu

Please also note the supplement to this comment:
http://www.solid-earth-discuss.net/se-2016-164/se-2016-164-RC2-supplement.pdf

[Figure]

**Supplement:**

[revised manuscript text omitted]

---

## Editor Comment (EC1) · R. Weinberg (Editor) · 9 Feb 2017

Dear Dr Kovaleva,

Thanks for submitting your manuscript to Solid Earth. I have now received two reviews. Both reviewers agree that this is a valuable contribution that should eventually be published. However, both agree in that the manuscript is poorly structured and that it requires major revision to get the message across succinctly and clearly. I ask you to revise the manuscript avoiding all repetition and that you restructure the discussion according to Rev 2. In particular describe the events in linear progression from the

early stages to the late stages. This amounts to a major revision.

The reviewers provided detailed comments directly on the pdf copies of the manuscript and this should help you in producing a final version. I am convinced that by removing repetition and chaotic structure, this paper could achieve a lot more. Some times less is more. Please go carefully through the reviewers comments and resubmit a much shorter article. At that point I will resend it to review, as per the suggestions provided.

Many thanks Roberto Weinberg
* * *

---

## Author Comment (AC1) · 6 Mar 2017

**N. Kelly (Referee)** nigel.kelly@colorado.edu

Overview

The paper authored by Kovaleva and others presents a detailed description of metamorphic zircon textures in metapelitic granulites from the Ivrea-Verbano Zone, northern Italy. The authors suggest, on the basis of textural relationships, but limited geochemical data, that the zircon formed as a result of hydration reactions, where Zr was liberated from ilmenite and later from Zr-rich rutile during prograde metamorphism and then cooling after peak granulite facies metamorphism.

The textures presented are an intriguing record of zircon formation during solid-state metamorphic reactions, and this work represents a valuable contribution to our understanding of zircon growth mechanisms. However, the current structure of the paper and how observations have been presented make it difficult to properly evaluate the veracity of the interpretations made and conclusions drawn. Direct criticisms and suggestions for improving the presentation of the paper are made below, with further comments made on an annotated PDF version of the manuscript.

General Comments

* Introduction
Much has been published on the mechanisms of zircon formation during metamorphism, a brief presentation of which would be very valuable in this contribution. The paper does cover some of the basics of "corona textures" and reports of these in metamorphic rocks, in particular meta-basic rocks. Although the focus on metabasites is curious, given this paper is on metapelites.
**In the overview we cite all reports on zircon corona (fine-grained) textures, both in metapelites (e.g. Pipe et al., 2016; Bingen et al., 2001; Ewing et al., 2013) and in metabasites (Söderlund et al., 2004; Charlier et al., 2007; Austrheim et al., 2008). They equally occur in metapelites and metabasites.**
However, it would be useful to see a recap of what we known about metamorphic zircon growth in all rock types as this does inform about the current study. In particular, this should cover differences between growth during major/accessory mineral breakdown, exsolution from accessory minerals, and precipitation from a fluid (vs growth as a result of major mineral decomposition during a retrograde hydration event). Some of this could be introduced during the discussion, but this background places the textures reported here in a clearer context. In this fashion, sections 1.1 and 1.2 could basically be integrated. **We have re-structured introduction section and added a subsection about zircon metamorphic growth (1.1 Growth of metamorphic zircon). Here we discussed zircon precipitation from fluid and melt, zircon growth as a result of breakdown of Zr-bearing minerals and as a result of exsolution from Zr-bearing minerals.**

* Geologic background & sample description
The description of the outcrop and interpretations of the origin of the rocks needs a clearer description. I would suggest a layout similar to: 1) mesoscopic outcrop features 2) mineralogy of the rocks 3) pertinent textures 4) interpretation of metamorphic features (PT-conditions, PT-path) including integration with published info.
**This section was restructured accordingly.**
A more detailed description of the sample taken would come later (see comments below). Some other points - the stronalite (an outdated term that could do with replacing **what would be the modern term to replace with?**) has been interpreted by others to have formed through partial

melting (not just "dehydration" during metamorphism). The leucosome in that rock may not just be injection of melt but melt formed in the rock itself. **This is noted and acknowledged. The term injection is removed, but we mentioned partial melting.**

But what I find most curious is how a layer of metasediment can be discordant to layering in the host stronalite. Can this be explained better? **I think it is not discordant, because this would be contra-intuitive and hard to imagine how this would form (may be tectonic contact, but the layer is rather thin). It probably just looked so because of the faulting in the host stronalite, with one set of faults normal to the sampled layer. The stronalite actually looks massive (besides of the faults). Thus our initial interpretation is probably wrong and we have removed it.**

But the focus here should really be to say that the rock studied has likely lost partial melt, so is somewhat restitic. **This is emphasized.** And then later it is cut by brittle faults. Somewhere in there ductile deformation affected the sample, but from the outcrop description is this not by any means given context. **This is mentioned.**

The mineralogy and textural description of the sample itself (page 4, line 34 onward) should be moved and inserted into the results section (4.1), as this first introduces the zircon occurrences and should be integrated with the rest of the textural descriptions. **This part was removed and placed in the beginning of the section 4 (new subsection 4.1).**

When the sample description (separate from outcrop description) is moved into section 4.1, there needs to be a clear and succinct description of the sample's major mineral textures. It is difficult to understand the relationships between prograde mineral textures (if present), peak mineral relationships, post-peak and hydration reaction textures (including the veins). From the description, I can't tell if there is more than one vein type – one that is dominated by Fe-Ti oxide minerals, the other dominated by fine-grained phyllosilicates, or are all a mixture of these? **This is clarified in the sample description section.** And the relationship of veins to mesoscopic and microscopic structures needs to be clearer – how can one deduce that a veins in thin-section "stretch mostly subvertically"? Is this an inference made from an oriented thin section, or just that when you look at the thin section it runs up and down? **In oriented thin section. They actually form conjugated (almost orthogonal) network. We have corrected and expanded this description. It must be clearer now.**

* Zircon textures (Results)
This needs to be more systematic. At present, zircon textures are presented as different "occurrences" that have much overlap. This reads like they came from different veins or thin sections and so were considered separately. I see an advantage to integrating the descriptions into "types" of textures.
For example:
" Zircon occurs in 2 textural contexts (as 2 textural types). The first, : : : The second, : : :"
**Introduced. We have subdivided the section according to the textural types.**
I see these types as the "vermicular" zircon (the coarser-grained zircon) and then the fine-grained "coronae" that occur either intergrown with Fe-Ti oxides mineral or in fractures.
These may occur with different "vein types" but these are the basic textural types.
**This is so! We have re-structured the data section accordingly.**

* Microprobe data
It is difficult to see, in its current form, what this contributes to the overall understanding of the problem. Very little is made of the data (apart from showing some compositional zoning in garnet). **We discussed change in composition of the main phases, calculated some geothermometers and noted that trace elements in ilmenite are more abundant then in rutile. Oh well, this is more of a structural and textural work, and limited chemistry we have just serves to support the main ideas.**
Nothing is made of the analysis of the phyllosilicates. **I am not sure – What can we make of**

**phyllosilicates in this regard?..** But perhaps most striking, is that zirconium hasn't been analyzed in rutile (or ilmenite). **Unfortunately, our microprobe analyses do not show Zr, it was apparently excluded from the analysed elements during the session!** This would be very illuminating! **Perhaps, but this is the data we got from a microprobe…** This would lend credence to the interpretation that Zr was probably sourced from ilmenite and rutile, and could also be used to estimate the temperatures of formation. Zr-in-Rt is very doable by electron microprobe. **This would be interesting, but such thermometry was already done and challenged (Ewing et al., 2013; Pape et al., 2016) and it's really slightly beyond the scopes of the current study, which aims to report textures.**

It would be beneficial to have Ti-in-zircon, although I understand with the fine-grained nature of the textures this would be impractical.
**I agree. Such data would not be very reliable.**
* Mineral reactions
The cool thing about textures such as those presented here, is that they do provide insights into the reactions that could have led to growth of zircon during metamorphism. **This is introduced.** However, this is weakly dealt with in this paper.

First – reaction 1 makes no sense. **Reaction 1 is removed.**

The metapelitic rocks have been shown to have experienced partial melting. Given the assemblage, I would suggest the reaction:
Bt + Sil + Qtz = Grt + Kfs + Qtz + Melt

This is not a "hydration" reaction, but a dehydration reaction, as water goes into the melt. This is also a prograde to peak reaction.
**I am not sure about this reaction, because: melt should have been separated from stronalite, which is much more felsic; garnet and sillimanite are clearly in equilibrium growth with each other, so its unlikely that garnet is a product of sillimanite resorption. Also, quartz is observed only in veins, so I think it came later.**
Formation of the veins, which host biotite **(no, the veins host chlorite and phengite),** and the development of elevated Fe and Mn in garnet rims, likely occurred during retrograde hydration. Resorption of garnet during retrograde breakdown has been commonly reported as a mechanism to elevate Fe and Mn in rims. **This is noted and corrected.** But, this mechanism MUST post-date the peak of metamorphism. **In this case garnet-biotite thermometry does not make much sense. Rims, which result in high-temperature estimations, are actually just a product of retrograde hydration and thus Fe enrichment.** While one can accept that the rock could have been hydrated from an external source and so could theoretically form during prograde metamorphism, the description of the mineralogy and textures does not support this. If you disagree with this interpretation, there must be an argument presented for it, and it is not present in the manuscript.
**We agree with your points and corrected the interpretations accordingly.**
There also needs to be a more developed discussion of the mineral relationships in the veins. For example, rutile is interpreted to have formed from the breakdown of ilmenite, which still occurs as relics in the veins, which (at least some of) are dominated by rutile-quartz intergrowths. What was the mineralogical make-up of the veins before reaction?
**Probably, ilmenite and quartz, these are the only relic phases that we observe.**
Does ilmenite (or rutile for that matter) occur elsewhere in the rock, or just confined to veins? **It occurs in veins and pockets between garnet and sillimanite.** Can you mass balance any reaction (even with a thought to open system**)? I am afraid, not, with the Fe-Ti oxides and unknown amount of melt, which was removed.** Did Ti come in with H2O and SiO2? What is the distribution of new zircon relative to the different vein types – is there less zircon when less rutile (and ilmenite)? **I don't think we have enough statistical data to make such suggestions, and from what we see here, zircon occurrence is random.**

**We have re-structured this section according to the reviewer suggestions.**
* Zircon textures
Leading on from the above discussion, are questions about whether fine-grained zircon formed from solid-state mineral breakdown (rutile, ilmenite) or due to exsolution of Zr from these minerals. The assumption in the paper is that this is just mineral breakdown-growth. However, many papers have suggested fine-grained textures represent exsolution during cooling.
**Textural evidence suggest breakdown. We do not exclude exsolution entirely, and we acknowledge such a possibility, but here will stick to breakdown interpretation.** Without a clear understanding of the veins themselves, I can only speculate, but it seems to me that a viable option for the formation of these textures involves the coarser zircon grains forming at high-T. They could have formed in the presence of crystallizing melt (e.g., zircon in Fig. 5 is seen associated with Sil, or included in garnet).
**That is right for other (non-coronae) zircon textures that we described briefly and presented in figure 2. For example, for zircon rims intergrowths with sillimanite. But in this study we focus on corona textures, so do not discuss extensively other zircon growth. Zircon coronae grow in Fe-Ti oxides clusters, which are, in their turn, postdate sillimanite (fill fractures in it – e.g. Fig. 1d). So I think that zircon growth with sillimanite and zircon growth in Fe-Ti oxides are not related and happened at a different stages.**
With cooling, Zr was less compatitible in rutile, so began to exsolve, hence the close textural relationship with rutile, and the lower-T not allowing grain coarsening. **We agree, thicker aggregates were formed at higher temperatures.** No doubt presence of fluids aided Zr mobility – hence presence along fractures and major grain boundaries with coarser-grained garnet or quartz. But evidence for pure metasomatic-driven reactions is not demonstrated here. **We agree, fluid was only responsible for slight local mobilization of Zr. We cleaned up our terminology and removed the term "metasomatism".**

Some comments on figures:
* there are some wonderful textures here. However, they are a little too small to see in some cases. Consider re-arranging and increasing the size so the relationships are clearly visible.
**We re-arranged and enlarged some of the figures, however, corona textures are so thin that even in enlarged figures it takes some effort to see them.**
* be sure to define all abbreviations – what is Phyl? **Defined in captions to Fig.1**
* there needs to be some clearer photomicrographs that show the metamorphic textures and the relationship to veins. If you want to push the idea of fluid-driven reaction, it would also help to present a systematic description (in the images) of the veins: the hydrous veins, rt-qtz intergrowths, etc: : : The current images make this very difficult to evaluate.
**We agree! Optical microscope images are added with clarification of described textures.**
* while schematic figures are excellent to explain the processes invoked, it would help if they properly described the textural development. In figure 6, the process starts with a single grain of ilmenite, which reacts to form a single grain of rutile, **well singe grain is just a schematic representation,** which reacts to for rutile-quartz aggregates. I don't recall this being described in the text. And in any case, the paper presents these textures as occurring in veins. I am left confused about the process here. **Why ilmenite grain cannot be sitting in veins or in pocket between peak metamorphic minerals? I am not sure where the contradiction to our model is. Our interpretation is that ilmenite is breaking down to rutile and to "vermicular" zircon, then this rutile re-crystallizes into quartz-rutile intergrowths with zircon coronae. This sequence we were trying to present in the figure. May be its just not artistic enough?**

**Thank you for your comments. They were of a great help to improve our work.**

---

## Author Comment (AC2) · 6 Mar 2017

[revised manuscript text omitted]

---

## Author Comment (AC3) · 6 Mar 2017

The paper presents an interesting case of secondary zircon growth as a result of mineral reactions which liberate Zr during cooling and exhumation from high grade conditions.

Although the growth of zircon as a consequence of the break-down of ilmenite and rutile, or diffusion of Zr from these minerals during cooling has been described before, the present paper adds an important dimension documenting zircon formation during retrogression even at greenschist facies conditions.

Although one can follow the descriptions and discussions reasonably well, some parts are rather confused and there is much repetition.
**We have cleaned up the text from most of the repetitions (except where repeating of information is necessary to develop the narrative for the presented thought).**
Chapter 1 introduces the subjects and reviews the previous literature, but then much of this is repeated again in the second chapter, and then again in the discussion. **We tried to clean up the repetitions.** The latter needs a thorough restructuring, ideally discussing the changes in mineralogy, structures and zircon features and evolution in a logical time progression. **We have re-structured the discussion.** Here the discussion starts with the late events eventually getting to the early stages, and circling around and back several times. **We have fixed this problem – now everything is presented in logical time progression (as we suggest it).** Chapter 5.2 is mainly a lengthy repetition of what has been said before, with a number of contradictory statements added in. **Noted! Shortened and cleaned up.**

I have marked the file and added some questions and comments there.
**We have gone through the attached file and introduced all of your corrections. We answered and addressed all your comments.**
My suggestion is to do a serious restructuring and condensation of the paper, sharpening the logic and cutting out the repetitions. A slender paper about 1/3 in size should be a result which the readers will greatly appreciate.
**We have reduced the discussion where appropriate.**
Febr. 4, 2017 F. Corfu

**Thank you very much for your help!**

---

## Author Comment (AC4) · 6 Mar 2017

[revised manuscript text omitted]

---

## Author Comment (AC5) · 9 Mar 2017

Dear Prof. Weinberg,

Thank you for your consideration.

We have carefully addressed reviewers comments and suggestions. We have shortened the text (discussion section specifically), got rid of unsupported assumptions and sharpened the logic. The resulted edited manuscript has been submitted. Point-to-point response was submitted as answers to each of the reviewers comments in the interactive discussion.

Best wishes, Elizaveta Kovaleva